# Scrub typhus association with autoimmune biomarkers and clinical implications

**Hwanseung You[1], Jeongeun Song[1], Seonglyeong Kim[1], Sang-Min Oh[2,3], Joo-Hee Hwang[2,3], Jeong-Hwan Hwang[2,3], Wan-Hee Yoo[2,3], Yunjung Choi** **[2,3]☯\*, Chang-Seop Lee**[2,3]☯\*

1 Jeonbuk National University Medical School, Jeonju, Republic of Korea, 2 Department of Internal Medicine, Jeonbuk National University Medical School, Jeonju, Republic of Korea, 3 Research Institute of Clinical Medicine of Jeonbuk National University-Biomedical Research Institute of Jeonbuk National University Hospital, Jeonju, Republic of Korea

☯ These authors contributed equally to this work.
\* imcyj@jbnu.ac.kr (YC); lcsmd@jbnu.ac.kr (CSL)

## Abstract

### Background

Scrub typhus, a disease caused by *Orientia tsutsugamushi*, triggers systemic vasculitis and is prevalent in Eastern and Southern Asia. This study aimed to uncover the relationship between scrub typhus and autoimmune responses, focusing on antinuclear antibodies (ANAs) and the implications of elevated ANA titers during infection.

### Method

Data from a total of 139 patients diagnosed with scrub typhus and 30 healthy controls were retrospectively analyzed through serum samples to assess the levels of ANAs and related autoantibodies. The study further examined the temporal variation of these antibody titers in relation to post-symptom progression and evaluated their association with clinical and laboratory parameters.

### Results

The study results detail significant differences in autoimmune responses between scrub typhus patients and healthy controls. ANA titers above 1:80 were detected in 71.2% of patients; these titers were elevated in only 13.3% of the healthy controls. High-titer positivity (1:640 or higher) was observed exclusively in the scrub typhus group. The distribution of ANA titers revealed a progressive increase in mean ANA and double-stranded deoxyribonucleic acid Immunoglobulin M (anti-dsDNA IgM) titers as the duration after symptom onset increased, indicating an augmented immune response over time. This trend was observed along with a systematic elevation in median anti-dsDNA IgM titers, highlighting the dynamic nature of immune responses in scrub typhus infection.

### Conclusion

Our findings conclude a substantial autoimmunological reaction in scrub typhus patients, suggesting potential mimicry of autoimmune conditions like Systemic Lupus

**Data availability statement:** All relevant data are within the paper and its Supporting information files.

**Funding:** CSL was supported by Fund of Biomedical Research Institute, Jeonbuk National University Hospital and by the Basic Science Research Programs (NRF-2018R1D1A3B07049557) of the National Research Foundation of Korea, which is funded by the Ministry of Education. The funders had no role in study design, data collection and analysis, decision to publish, or preparation of the manuscript.

**Competing interests:** The authors have declared that no competing interests exist.

Erythematosus. These results underline the complex interplay between infectious diseases and autoimmunity, emphasizing the necessity for further research into these mechanisms and their clinical implications.

## Author summary

This study investigates the autoimmune responses in scrub typhus patients, finding significantly higher antinuclear antibodies and double-stranded deoxyribonucleic acid Immunoglobulin M titers compared to healthy controls, with these titers increasing over time post-symptom onset. It highlights the potential of scrub typhus to mimic autoimmune conditions like systemic lupus erythematosus, emphasizing the need for further research into the interaction between infectious diseases and autoimmunity for improved diagnosis and management.

## Introduction

Scrub typhus is an acute febrile illness caused by Rickettsia infection, specifically *Orientia tsutsugamushi* infection, transmitted through Leptotrombidium mites [1]. This disease primarily afflicts individuals in Eastern and Southern Asia, including Korea, China, Vietnam, and Taiwan, with a peak incidence during the autumn [2,3]. Its clinical spectrum varies widely, from constitutional manifestations to potentially fatal outcomes. Characteristic clinical features include fever, rash, eschar, and lymphadenopathy; however, patients may also present with acute respiratory distress syndrome and interstitial pneumonia. Sequelae affecting the cardiovascular and central nervous systems, underscoring the systemic nature of vasculitis induced by the infection, are not uncommon [4,5].

Antinuclear antibodies (ANAs) encompass a broad spectrum of autoantibodies that target intranuclear constituents, including both nucleic acids and proteins [6]. The detection of ANAs signals an aberrant immune response in which the body's defenses erroneously attack host cells. ANAs are, therefore, critical biomarkers for various autoimmune pathologies, notably systemic lupus erythematosus (SLE). Given its integral role in the identification, classification, and development of prognoses in such disorders, ANA testing is an essential preliminary diagnostic tool in the assessment of systemic autoimmune conditions [7].

Infection has been postulated to contribute to the development of autoimmunity, acting synergistically with genetic and environmental factors. In the context of scrub typhus, infection by *Orientia tsutsugamushi* targets endothelial cells lining the vascular system, precipitating vasculitis. Although the immunological response aims to neutralize the invading pathogen, this response inadvertently escalates vascular inflammation, exacerbating the condition. Concurrently, the exposure of autoantigens or the dysregulation of the immune system during this inflammatory response may catalyze the onset of autoimmunity. Alterations in the expression of ANAs among scrub typhus patients have been documented [8], emphasizing the dynamic nature of the immune response throughout the disease progression.

The primary objective of our study was to investigate whether scrub typhus, due to its vasculitic pathogenesis, could potentially trigger autoimmune responses. Specifically, we aimed to elucidate the development of autoimmunity by examining the expression of ANAs and associated autoantibodies, such as double-stranded deoxyribonucleic acid IgM (anti-dsDNA IgM), across the different stages of scrub typhus infection and to assess their clinical significance.

The rationale for measuring these autoantibodies in scrub typhus patients is based on the hypothesis that infections can act as environmental triggers for autoimmune responses. By evaluating the levels of these autoantibodies, we aimed to explore the potential link between scrub typhus infection and the early pathological processes of autoimmune diseases.

## Methods

### Ethics statement

This study was approved by the Institutional Review Board (IRB) of Jeonbuk National University Hospital, and all patients provided written informed consent (IRB registration number 2023-08-056).

### Patients and data collection

A retrospective study was conducted in a tertiary hospital in the southwestern Korean city of Jeonju between January 2016 and December 2022. The selection of patients with scrub typhus included those who were clinically suspected of having scrub typhus and were 18 years or older. Laboratory diagnosis of scrub typhus was based on one of these criteria: (1) An increase in indirect immunofluorescence assay (IFA) total antibody (IgM/G) titer ≥ 1:80 against *O. tsutsugamushi*, or (2) ≥ four-fold increase in IFA titer in paired sera, or (3) positive result from a nested polymerase chain reaction (PCR) targeting the 56-kDa type-specific antigen gene of *O. tsutsugamushi,* which refers to antigens that are specific to a particular strain or type of *O. tsutsugamushi* [9].

We utilized IFA, a serological assay endorsed by the Korea Disease Control and Prevention Agency (KDCA), to detect serum IgM and IgG antibodies in patients responding to *Orientia tsutsugamushi* antigens. In accordance with the KDCA protocol, human sera were subjected to two-fold serial dilutions, beginning at 1:40, and subsequently reacted with *O. tsutsugamushi* antigens [10]. The IFA tests are conducted at a specialized laboratory, using antigens from Boryong, Gilliam, and Karp strains to measure IgM/IgG (total antibodies). The results are interpreted by two or more readers to ensure accuracy.

In this study, patients who met the diagnostic criteria based on IFA were as follows: 1) IFA detected IgM/G (total antibodies) in 82 out of 139 patients (59.0%), 2) a four-fold or greater increase in IFA titer in paired sera was observed in 34 out of 139 patients (24.5%), and 3) PCR confirmed positivity in all 139 patients (100%). Healthy controls were recruited from individuals who visited the same hospital for routine health check-ups during the study period. The inclusion criteria for healthy controls were: (a) age 18 years or older, (b) no history of autoimmune diseases, and (c) no recent infections or chronic illnesses. These criteria were established to ensure that the control group represented a baseline ANA titer distribution without underlying conditions that could influence the results.

To address the potential confounding effect of age distribution differences between the control group and the scrub typhus group on ANA titers, we included age as a key variable in our analysis. The age range of patients with scrub typhus was from 22 to 89 years, while the age range of the healthy controls was from 20 to 65 years. Recognizing that ANA positivity is inherently higher in older individuals, we implemented the following measures:

Age Stratification: We stratified the study population into age groups to analyze the impact of age on ANA titers more precisely. Specifically, we conducted subgroup analyses focusing on patients aged 40–59 years. This age range was selected to ensure a sufficient sample size for statistical analysis while minimizing the age difference between groups.

Subgroup Analysis: We compared the ANA titers within these age-defined subgroups. The demographic and immunological characteristics of healthy controls and patients with scrub

typhus aged 40–59 years are presented in S1 Table. This analysis allowed us to evaluate the relationship between scrub typhus infection and ANA titers independently of age-related factors.

Statistical Adjustments: In addition to subgroup analyses, we applied statistical adjustments to account for age as a covariate in our analysis of ANA titers. By including age as a covariate in our regression models, we aimed to control for its potential confounding effect and isolate the impact of scrub typhus infection on ANA titers.

## DNA extraction from scrub typhus patients

Whole blood samples from scrub typhus patients were collected, and peripheral blood mononuclear cells (PBMC) were isolated using Lymphoprep density gradient medium and SepMate tubes (Stemcell Technologies, Vancouver, Canada). The isolated PBMC were aliquoted and stored at −80°C. After isolation, DNA was purified using the QIAamp DNA Mini Kit (QIAGEN GmbH, Hilden, Germany) according to the manufacturer's instructions. Purified DNA was aliquoted and stored at −20°C.

## DNA amplification and sequencing for bacterial identification

To confirm the presence of *O. tsutsugamushi*, a nested PCR targeting the 56-kDa gene of *O. tsutsugamushi* was performed. Primers 34 (forward, 5'-TCA AGC TTA TTG CTA GTG CAA TGT CTGC-3', the 56-kDa gene based on the Gilliam strain) and 55 (5'-AGG GAT CCC TGC TGC TGT GCT TGC TGCG-3') were used in the first PCR. Nested PCR primers 10 (5'-GAT CAA GCT TCC TCA GCC TAC TAT AAT GCC-3') and 11 (5'-CTA GGG ATC CCG ACA GAT GCA CTA TTA GGC-3') were used in the second PCR amplification to generate a 483-bp fragment. Nested PCR was performed as described by Lee et al. [11]. The amplified PCR products were confirmed by agarose gel electrophoresis and purified from the agarose gel using the QIAquick gel extraction kit (QIAGEN) and the Expin Gel SV kit (GeneAll, Seoul, Korea). Each PCR product was analyzed and confirmed by sequencing.

## Multilocus sequence typing (MLST) and data analyses

Genomic DNA of *O. tsutsugamushi* was characterized using MLST, a method of characterizing isolates of microbial species using the sequences of internal fragments of multiple (usually seven) housekeeping genes, as previously described [12]. Housekeeping genes (*gpsA, mdh, nrdB, nuoF, ppdK, sucB, sucD*) were amplified by PCR and sequenced in forward and reverse directions with their associated primers. The amplified PCR products were confirmed by agarose gel electrophoresis, purified, and sequenced [12,13].

## Autoantibody and laboratory testing

The ANA titers were determined using an indirect immunofluorescence assay (IFA) on HEp-2 cells. A titer of 1:80 or greater was deemed positive, reflecting potential autoimmune reactivity. In our study, antibody titer tests were conducted from the first day of symptom onset up to a maximum of 40 days. The follow-up periods were segmented into three distinct intervals: 1–3 days, 4–7 days, and beyond 7 days. This segmentation allowed us to analyze the variations in ANA and anti-dsDNA IgM titers based on the temporal progression post-symptom onset. The quantification of anti-dsDNA antibodies was performed utilizing an enzyme-linked immunosorbent assay (ELISA). Results were reported in units per milliliter (U/mL); the normal reference range was at or below 20 U/mL. Levels of C3 and C4 were measured by nephelometry. The reference ranges for these components are 90–180 mg/dL for C3 and 10–40 mg/dL for C4.

A series of routine laboratory tests was performed in a certified clinical laboratory under stringent quality control measures to ensure reliability and accuracy of the results.

## Statistical analyses

Descriptive statistics were expressed as numbers and percentages, means and ranges, or medians and interquartile ranges (IQRs). Statistical analyses were performed with IBM SPSS Statistics for Windows, version 25.0 (IBM Corp., Armonk, NY, USA). We have specified the types of statistical tests employed, including t-tests for comparing means between two groups, ANOVA for comparing means among three groups, and chi-square tests for examining the relationship between categorical variables. Additionally, we have detailed the use of univariate and multivariate logistic regression analyses. For the multivariate analysis, we included variables that had a *p*-value of up to 0.2 in the univariate analysis. The significance level for interpreting the *p*-values was set at 5%. We performed the Kruskal-Wallis test to identify overall group differences, followed by post-hoc pairwise comparisons using the Mann-Whitney U test. To control for the increased risk of Type I errors, we applied the Bonferroni correction, adjusting the significance level by dividing it by the number of comparisons.

## Results

### Elevated antinuclear antibody titers in patients with scrub typhus compared to healthy controls

In this study, we evaluated the ANA titers of patients diagnosed with scrub typhus compared to those of healthy controls. Data from a total of 139 cases of scrub typhus and 30 healthy control cases were analyzed using serum samples. Among the healthy controls, 13.3% exhibited ANA titers exceeding 1:80; however, a significant 71.2% of the scrub typhus group displayed ANA titers above this threshold (Table 1). As depicted in Fig 1, patients with scrub typhus demonstrated elevated ANA titers across several dilutions, including 1:80, 1:160, and 1:320. None of the healthy controls exhibited titers of 1:640 or 1:1,280, but 5% of the scrub typhus patients showed high-titer positivity at these levels.

Table 1. Demographic and Immunological Characteristics of Healthy Controls and Patients with Scrub Typhus.

| | Controls | Scrub Typhus | Total | *p* |
|---|---|---|---|---|
| | (N = 30) | (N = 139) | (N = 169) | |
| **Gender, n (%)** | | | | 0.505 |
| F | 18 (60.0) | 95 (68.3) | 113 (66.9) | |
| M | 12 (40.0) | 44 (31.7) | 56 (33.1) | |
| **Age** | 35.8 ± 15.0 | 67.3 ± 12.6 | 61.7 ± 17.8 | < 0.001 |
| **Anti-dsDNA IgM (U/mL)** | 7.2 ± 10.0 | 14.5 ± 14.4 | 13.2 ± 14.0 | 0.002 |
| **ANA titer, n (%)** | | | | < 0.001 |
| < 1:80 | 23 (76.7) | 40 (28.8) | 152 (54.7) | |
| 1:80 | 3 (10.0) | 47 (33.8) | 59 (21.2) | |
| 1:160 | 3 (10.0) | 22 (15.8) | 33 (11.9) | |
| 1:320 | 1 (3.3) | 23 (16.5) | 25 (9.0) | |
| 1:640 | 0 (0.0) | 5 (3.6) | 5 (1.8) | |
| 1:1,280 | 0 (0.0) | 2 (1.4) | 2 (1.2) | |

ANA, anti-nuclear antibody; Anti-dsDNA IgM, Anti-double-stranded DNA IgM. Data are presented as mean ± S.D. or number (percentage).

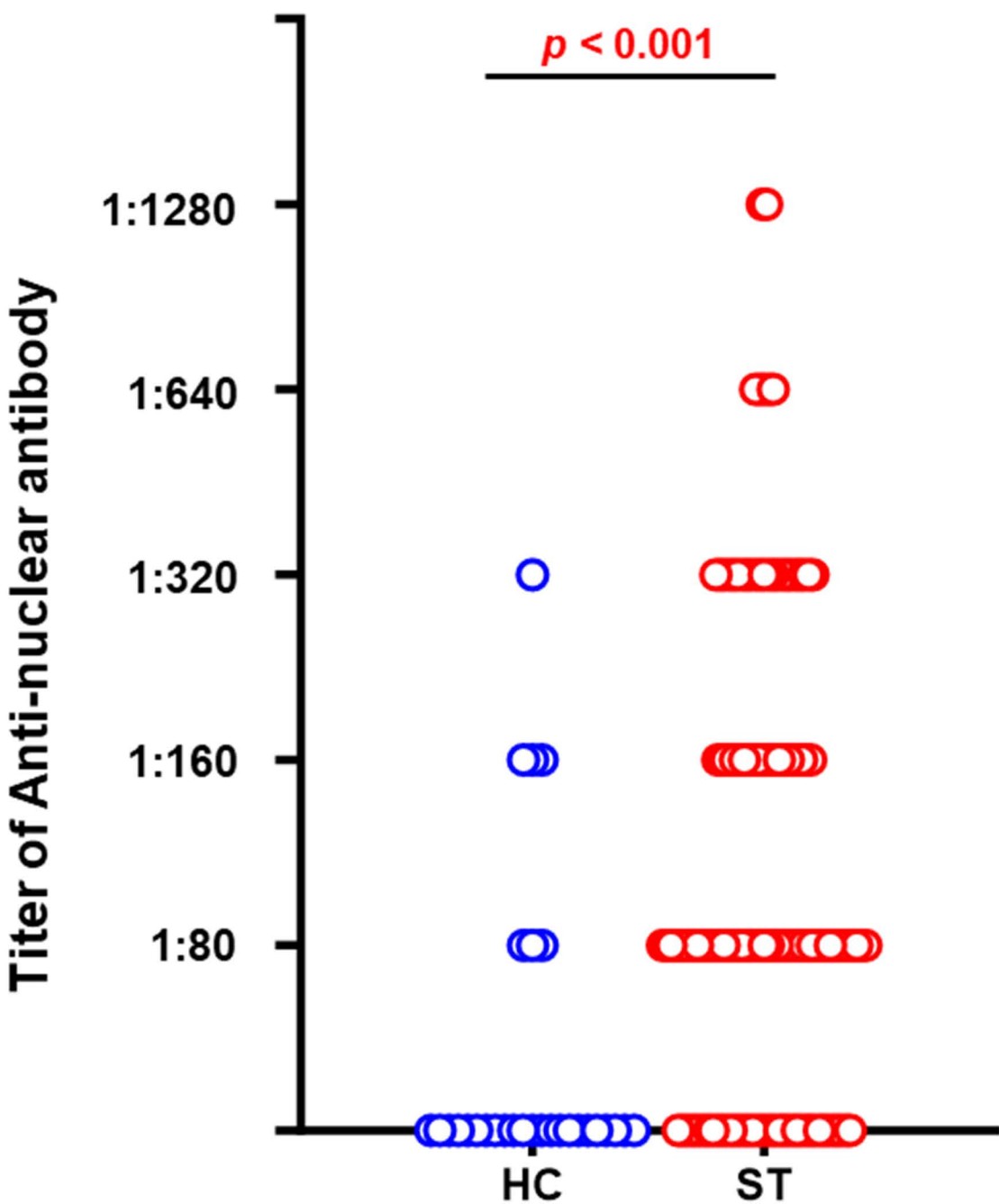

**Fig 1. Comparison of ANA titer between healthy controls and patients with scrub typhus.** ANA titer levels among healthy controls (blue) compared to patients with scrub typhus (red). The titer values are shown at increasing dilutions; each dot represents an individual's titer measurement. *p* < 0.001 compared to the healthy control group.

### Elevated anti-dsDNA antibody titers in patients with scrub typhus compared to healthy controls

The ANA test is a highly sensitive screening tool for systemic lupus erythematosus (SLE), and a positive ANA result is one criterion for SLE diagnosis according to the 2019 European

League Against Rheumatism/American College of Rheumatology (EULAR/ACR) classification criteria [14]. Our study extended its analysis to include SLE-specific anti-double-stranded DNA (anti-dsDNA) antibody titers. We compared these titers between patients diagnosed with scrub typhus and healthy control subjects. The findings, as presented in Table 1, indicate a significant elevation in the mean anti-dsDNA IgM titer among the scrub typhus patients (14.5 ± 14.4 U/mL) compared to the healthy controls (7.2 ± 10.0 U/mL, $p$ = 0.002). Given that the reference range for anti-dsDNA IgM is below 20 U/mL, our analysis further revealed a higher prevalence of positive anti-dsDNA IgM results in the scrub typhus group, as shown in Fig 2.

## Temporal variation of ANA and anti-dsDNA IgM titers in scrub typhus patients

Building upon the higher ANA and anti-double-stranded DNA IgM titers in the scrub typhus group compared to the healthy control group, we conducted a further analysis on the variations in these antibody titers based on the temporal progression post-symptom onset. The analysis segmented the duration from symptom onset to blood sampling into three distinct intervals: 1–3 days, 4–7 days, and beyond 7 days. One finding was that of a progressive increase in the mean titer of ANA in patients with scrub typhus as the duration from symptom onset increased. The most significant elevations in ANA titers were observed after the first week of symptom onset (Fig 3). This change was significant; both the positivity rate and the magnitude of ANA titers, which varied from 1:80 to 1:1,280, were elevated. These data underscore a marked augmentation in immune response intensity correlating with the elapsed time from symptom onset. The titer of anti-dsDNA IgM demonstrated a pattern congruent with that of the ANA titers; the apex value occurred more than seven days from the onset of symptoms. These parallel findings for ANA and anti-dsDNA IgM titers underscore the consistency of the immune response in scrub typhus patients. The observed increase in anti-dsDNA IgM titers during this timeframe was also significant (Fig 4).

Table 2 delineates median titers, revealing a systematic escalation in the median titers of anti-dsDNA IgM to 6.4 U/ml, 8.7 U/ml, and 12.1 U/ml, sequentially aligned with the progression of time from symptom onset. This change was statistically significant. The median intervals from symptom onset to the collection of samples were 2.5 days, 6.0 days, and 11.0 days, respectively. Demographic data analysis revealed no substantial gender or age variance across the three groups. In the context of complement components 3 (C3) and 4 (C4), which are markers of disease activity in SLE, a notable contrast was observed. While the levels of C3 remained consistent across the evaluated groups, C4 levels exhibited a discernible decline over time.

## Association of antinuclear antibody titers with clinical and laboratory parameters in scrub typhus patients

The observed higher ANA positivity rate and titer in patients diagnosed with scrub typhus compared to a healthy control group, coupled with variations in ANA titer within the scrub typhus group based on the timeframe since symptom onset, led to the focus of our study. We aimed to uncover potential clinical or laboratory differences that correlate with the ANA titer levels in scrub typhus patients. Accordingly, patients were divided into three groups based on ANA titer: < 1:80, ≥ 1:80 and < 1:320, and ≥ 1:320. This classification allowed a detailed examination (Table 3).

We performed logistic regression analyses to evaluate the independent relationship between ANA titers and various clinical and laboratory parameters. The analysis included adjustments for potential confounders such as age and sex. The results of these analyses are

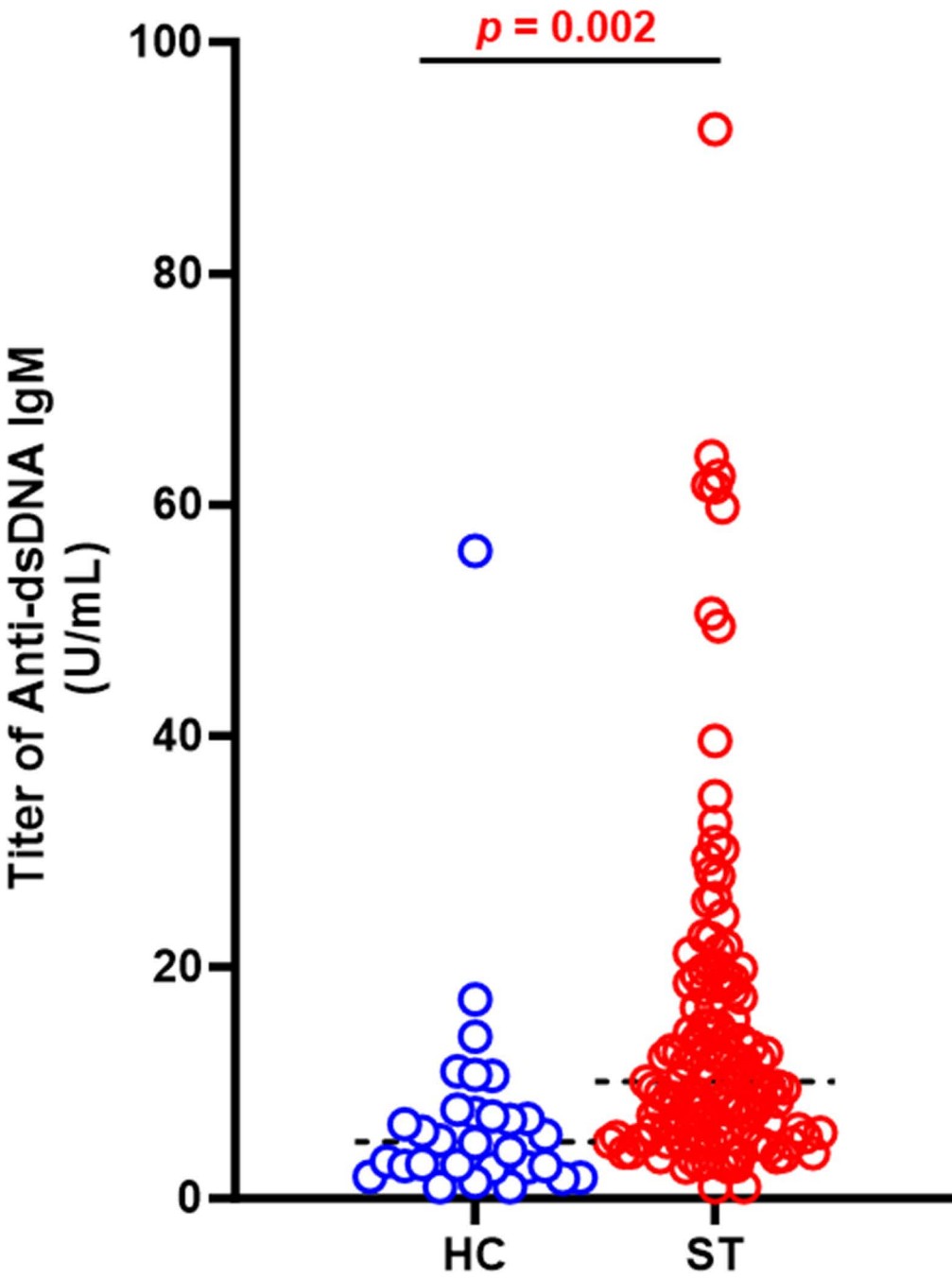

**Fig 2. Comparison of anti-dsDNA antibody titer between healthy controls and scrub typhus patients.** The scatter plot contrasts anti-dsDNA antibody titers between healthy controls (blue dots) and scrub typhus patients (red dots). $p = 0.002$ compared to the healthy control group.

summarized S2–S5 Tables. The analysis revealed significant variances of several parameters across the groups, including patient age; the interval from symptom onset to blood sampling; and levels of anti-dsDNA IgM (S6 Table), C4, and serum albumin. The average age within these groups was primarily between 60 and 69 years, with a discernible trend toward higher

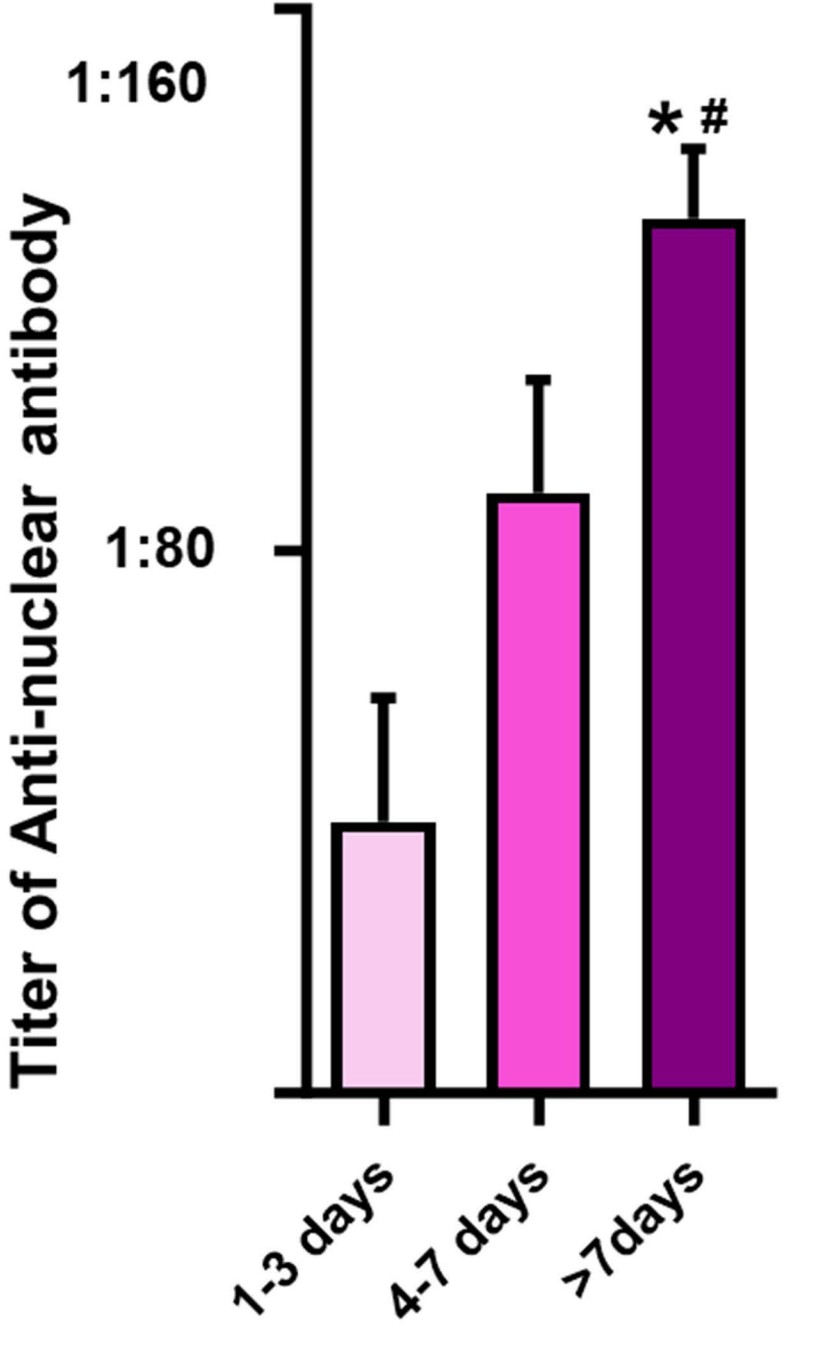

**Fig 3. Temporal variation of ANA titers post symptom onset in scrub typhus patients.** This bar graph presents the progression of ANA titers in patients diagnosed with scrub typhus categorized by the number of days since the onset of symptoms. Data are presented as mean ± standard error of the mean (SEM). * *p < 0.001* compared to the 1–3 days group; # *p* = 0.018 compared to the 4–7 days group.

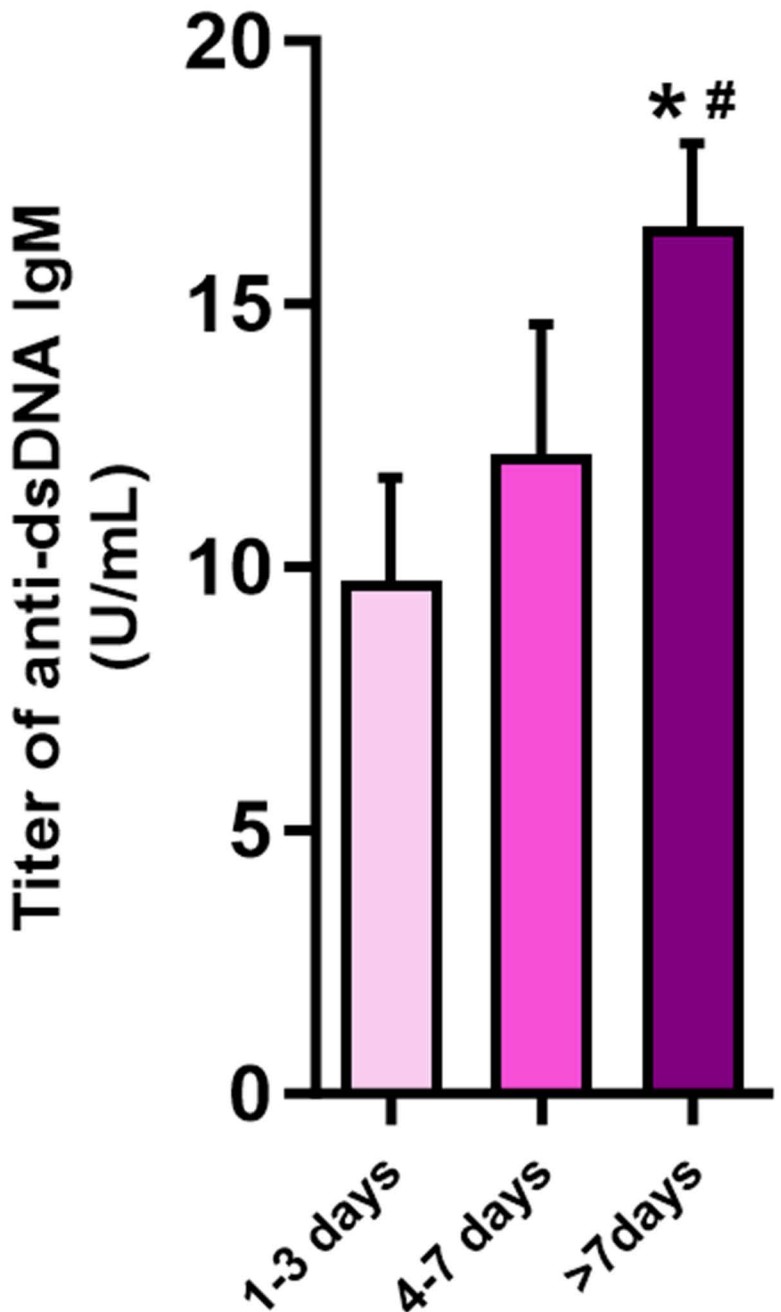

**Fig 4. Temporal analysis of anti-dsDNA antibody titers following symptom onset in scrub typhus.** The bars depict a quantitative assessment of anti-dsDNA antibody titers in patients with scrub typhus categorized by the time elapsed since the onset of symptoms. Data are presented as mean ± SEM. * $p = 0.045$ compared to the 1–3 days group; # $p = 0.025$ compared to the 4–7 days group.

average age in the group characterized by elevated antibody titers. Additionally, this group was marked by extended durations from symptom onset to blood sampling and exhibited

**Table 2. Clinical Characteristics and Autoantibody Profiles Stratified by Symptom Onset to Blood Sampling Duration in Scrub Typhus Patients.**

| Onset to sampling duration | 1–3 days | 4–7 days | > 7days | p |
|---|---|---|---|---|
| | (N = 14) | (N = 37) | (N = 88) | |
| **Gender, n (%)** | | | | 0.485 |
| F | 8 (57.1) | 24 (64.9) | 63 (71.6) | |
| M | 6 (42.9) | 13 (35.1) | 25 (28.4) | |
| **Age (years)** | 70.0 [63.0–79.0] | 68.0 [60.0–76.0] | 68.5 [60.5–77.5] | 0.792 |
| **Symptom onset-to-blood sampling duration (days)** | 2.5 [2.0–3.0] | 6.0 [5.0–6.0] | 11.0 [9.0–16.0] | < 0.001 |
| **ANA titer, n (%)** | | | | 0.019 |
| < 1:80 | 9 (64.3) | 16 (43.2) | 15 (17.0) | |
| 1:80 | 4 (28.6) | 10 (27.0) | 33 (37.5) | |
| 1:160 | 0 (0.0) | 4 (10.8) | 18 (20.5) | |
| 1:320 | 1 (7.1) | 5 (13.5) | 17 (19.3) | |
| 1:640 | 0 (0.0) | 2 (5.4) | 3 (3.4) | |
| 1:1,280 | 0 (0.0) | 0 (0.0) | 2 (2.3) | |
| **Anti-dsDNA IgM (U/mL)** | 6.4 [3.8–18.4] | 8.7 [5.8–12.2] | 12.1 [7.0–19.2] | 0.026 |
| **C3 (mg/dL)** | 117.5 [94.0–147.0] | 143.0 [127.0–152.0] | 125.0 [105.0–142.5] | 0.063 |
| **C4 (mg/dL)** | 34.5 [25.0–38.0] | 34.0 [27.0–41.0] | 24.0 [18.0–35.0] | 0.003 |

ADA, adenosine deaminase; ANA, anti-nuclear antibody; Anti-dsDNA IgM, anti-double-stranded DNA IgM; C3, complement 3; C4, complement 4, Data are presented as median [interquartile range] or number (percentage).

elevated anti-dsDNA IgM and diminished C4 levels. These patterns not only corroborate the observations presented in Table 2, which analyzes data in relation to the timeline of symptom onset, but also substantiates the relationship between higher ANA titers and specific clinical manifestations. In the chemistry analysis, no statistically significant differences were observed among the groups under study. However, a trend toward a serum albumin level decrease with a concomitant C-reactive protein and white blood cell count increase was observed.

## Discussion

In the present investigation, elevated antinuclear antibodies (ANA) and anti-double stranded DNA (anti-dsDNA) IgM titers were observed in patients diagnosed with scrub typhus in comparison to a control group comprising healthy individuals. Augmentation in the titers of these antibodies was identified as the disease progressed to its later stages. Upon analyzing the clinical and laboratory findings in correlation with the titration of ANA, we determined that there were no significant differences among the groups.

Consistent with other reported findings [8], our study identified an increased ANA frequency and titer in the group of patients with scrub typhus. This observation aligns with demonstrations that infections can act as triggers for autoimmune responses, offering a potential explanation for the elevated ANA levels observed in these patients. We observed an age difference between the control group and the scrub typhus group; the scrub typhus group tended to be older. Nevertheless, the distribution of ANA titers cannot be solely attributed to age. Previously conducted studies have indicated that approximately 3% of healthy controls have ANA titers ≥ 1:320, and that 20.63% of healthy elderly Koreans demonstrate ANA positivity at titers ranging from 1:160 to 1:1,280 [15,16]. Our study revealed that 21.5% of scrub typhus patients had ANA titers ≥ 1:320, and 37.4% showed positivity at titers ranging from

**Table 3. Stratification of Clinical and Laboratory Parameters in Scrub Typhus Patients by ANA Titer Intervals.**

| | < 1:80 | 1:80, < 1:320 | ≧ 1:320 | Total | P |
|---|---|---|---|---|---|
| | (N = 40) | (N = 47) | (N = 52) | (N = 139) | |
| **Gender** | | | | | |
| Male, n (%) | 10 (25.0) | 16 (34.0) | 18 (34.6) | 44 (31.7) | 0.562 |
| **Age (years)** | 63.1 ± 15.6 | 68.3 ± 12.0 | 69.7 ± 9.7 | 67.3 ± 12.6 | 0.037 |
| **Symptom onset-to-blood sampling duration (days)** | 7.0 ± 4.5 | 10.9 ± 7.8 | 12.3 ± 7.0 | 10.3 ± 7.0 | 0.001 |
| **Anti-dsDNA IgM (U/mL)** | 8.7 ± 5.8 | 14.4 ± 15.0 | 19.3 ± 16.8 | 14.6 ± 14.5 | 0.002 |
| **C3 (mg/dL)** | 135.6 ± 30.1 | 127.3 ± 31.2 | 122.4 ± 27.5 | 127.8 ± 29.8 | 0.109 |
| **C4 (mg/dL)** | 35.8 ± 13.7 | 29.0 ± 12.5 | 26.3 ± 9.9 | 29.9 ± 12.5 | 0.001 |
| **Admission duration (days)** | 5.6 ± 2.3 | 7.0 ± 4.2 | 7.1 ± 3.6 | 6.5 ± 3.5 | 0.153 |
| **Diabetes** | 6 (18.2) | 9 (25.7) | 3 (10.3) | 18 (18.6) | 0.289 |
| **Hypertension** | 14 (42.4) | 17 (48.6) | 12 (41.4) | 43 (44.3) | 0.816 |
| **Genotype, n (%)** | | | | | 0.565 |
| Boryong | 9 (22.5) | 12 (25.5) | 8 (15.4) | 29 (20.9) | |
| Karp | 1 (2.5) | 2 (4.3) | 1 (1.9) | 4 (2.9) | |
| Kawasaki | 1 (2.5) | 0 (0.0) | 0 (0.0) | 1 (0.7) | |
| **Fever, n (%)** | 32 (97.0) | 34 (97.1) | 28 (96.6) | 94 (96.9) | 0.990 |
| **Headache, n (%)** | 12 (36.4) | 16 (45.7) | 10 (34.5) | 38 (39.2) | 0.605 |
| **Loss of appetite, n (%)** | 5 (15.2) | 12 (34.3) | 6 (20.7) | 23 (23.7) | 0.161 |
| **Nausea/Vomiting, n (%)** | 11 (33.3) | 12 (34.3) | 7 (24.1) | 30 (30.9) | 0.638 |
| **Abdominal pain, n (%)** | 7 (21.2) | 6 (17.1) | 9 (31.0) | 22 (22.7) | 0.405 |
| **Rash, n (%)** | 21 (63.6) | 21 (60.0) | 23 (79.3) | 65 (67.0) | 0.231 |
| **Eschars, n (%)** | 24 (72.7) | 30 (85.7) | 25 (86.2) | 79 (81.4) | 0.284 |
| **WBC, ×1,000/mm³** | 6.5 ± 5.6 | 8.3 ± 3.7 | 9.1 ± 3.9 | 7.9 ± 4.6 | 0.071 |
| **PLT, ×1,000/mm³** | 126.2 ± 54.0 | 132.2 ± 50.8 | 119.4 ± 47.8 | 126.3 ± 50.5 | 0.607 |
| **hs-CRP, mg/dL** | 92.9 ± 68.6 | 114.9 ± 68.3 | 117.8 ± 53.8 | 108.3 ± 64.7 | 0.242 |
| **AST, IU/L** | 135.4 ± 132.9 | 106.0 ± 73.2 | 93.4 ± 48.8 | 112.2 ± 93.7 | 0.189 |
| **ALT, IU/L** | 98.6 ± 95.7 | 79.3 ± 57.7 | 73.8 ± 62.3 | 84.2 ± 74.0 | 0.377 |
| **ALP, IU/L** | 135.2 ± 95.9 | 127.1 ± 64.7 | 134.6 ± 108.7 | 132.1 ± 89.5 | 0.920 |
| **Total bilirubin, mg/dL** | 1.0 ± 1.1 | 0.9 ± 0.7 | 0.7 ± 0.3 | 0.9 ± 0.8 | 0.399 |
| **Albumin, g/dL** | 3.8 ± 0.5 | 3.5 ± 0.5 | 3.4 ± 0.5 | 3.6 ± 0.5 | 0.001 |
| **Creatinine, mg/dL** | 1.0 ± 0.7 | 1.0 ± 0.6 | 1.1 ± 0.8 | 1.0 ± 0.7 | 0.850 |
| **PT, INR** | 1.1 ± 0.1 | 1.1 ± 0.3 | 1.4 ± 1.3 | 1.2 ± 0.8 | 0.195 |

ALP, alkaline phosphatase; ALT, alanine aminotransferase; ANA, anti-nuclear antibody; anti-dsDNA IgM, anti-double stranded DNA IgM; AST, aspartate aminotransferase; C3, complement 3; C4, complement 4; hs-CRP, high-sensitivity C-reactive protein; PT, prothrombin time. Data are presented as mean ± S.D. or number (percentage).

1:160 to 1:1,280. This suggests that the elevated ANA titers observed in patients with scrub typhus cannot be solely attributed to aging.

SLE is a classic autoimmune disorder, with elevated ANA titers serving as a central marker for its evaluation. In our study, we assessed both ANA titers and anti-dsDNA antibodies, which are highly specific for diagnosing SLE. Given that infections can act as environmental triggers for autoimmunity [17], our findings led us to explore the potential association between scrub typhus and SLE. Our results suggest that scrub typhus infection could potentially trigger the early pathological course of SLE. This potential link may be explained by the release of sequestered antigens into the circulation due to cell and tissue

damage during the vasculitic phase of scrub typhus. Such antigen release could stimulate an autoimmune response, resulting in elevated levels of ANA and anti-dsDNA antibodies. Furthermore, the clinical manifestations of scrub typhus, such as fever, anemia, and thrombocytopenia, mimic those seen in SLE, supporting the hypothesis that scrub typhus may trigger autoimmune processes in genetically predisposed individuals. We observed an increase in anti-dsDNA antibodies and a decrease in C4 levels during the late stages of scrub typhus infection, which align with the immune response patterns seen in SLE. This observation is supported by two main factors: the release of sequestered antigens into the circulation following cell and tissue damage during vasculitis [18], and the clinical similarities between scrub typhus and SLE, including fever, anemia, and thrombocytopenia. These similarities pose diagnostic challenges for clinicians until distinct evidence, such as the presence of an eschar, is discovered. This clinical and immunological overlap raises intriguing questions about the potential for infections to precipitate or exacerbate autoimmune responses in predisposed individuals.

Therefore, we conducted an analysis of antibody titer trends over different stages of scrub typhus infection. This analysis detects nuances in the immunological response throughout the infection. Late-stage anti-dsDNA IgM and C4 levels showed significant changes resembling those of SLE, while C3 levels remained consistent throughout the infection process. This divergence emphasizes the unique immunological and disease activity marker responses in scrub typhus, suggesting a complex interaction of immune mechanisms. Immune pathway activation in scrub typhus immunopathogenesis is distinct and may mimic autoimmune conditions such as SLE.

Our next investigation focused on elucidating the correlation between scrub typhus and autoimmunity; we assessed laboratory and clinical manifestations in patients diagnosed with scrub typhus and ANA titers in testing this correlation. Our hypothesis was that ANA titer can serve as a marker for distinct scrub typhus disease attributes or immunological responses. Findings revealed an increase in anti-dsDNA antibodies and a decrease in C4 levels consistent with classifications based on disease duration. Infections are known to induce vasculitis through mechanisms such as immune cell-mediated inflammation and direct endothelial damage [19]. During the course of infection, the immune system responds aggressively, leading to localized inflammation of the blood vessels. This inflammation can result in endothelial injury, which in turn may expose immune cells to self-antigens [20]. As a consequence, the prolonged activation of the immune system could initiate the production of autoantibodies, linking the initial infection to the development of autoimmunity. In the case of scrub typhus, it is plausible that the persistent immune activation observed during infection contributes to the generation of autoantibodies, further supporting the hypothesis that infections can serve as a trigger for autoimmune diseases. This potential link between infection-induced vasculitis and subsequent autoantibody production underscores the importance of exploring autoimmune mechanisms in the context of infectious diseases. The observed changes suggest that autoimmunity intensifies in later stages of scrub typhus and reinforce the hypothesized correlation between ANA titers and disease dynamics. In scrub typhus patients, serum albumin levels fall and ANA titers rise. While these changes may not significantly deviate from the normal range, a subtle interplay between ANA titers and patient physiology is suggested. Inflammation progression during the natural course of scrub typhus infection can be tracked through levels of albumin, a marker of inflammation [21], and ANA titer. This titer reflects disease progression; ANA titer changes across the disease progression timeline are relatively consistent. C-reactive protein (CRP) levels or white blood cell (WBC) counts may also indicate the course of inflammation, but these changes may not be statistically significant.

Nutritional deficiencies may develop during scrub typhus infection. The combination of the effects of prolonged inflammation and nutritional inadequacies underline the complexity of scrub typhus pathogenesis and its interaction with host immune responses, an interaction marked by variations in ANA titers.

The current study has limitations. The numbers of cases and healthy controls were limited. Despite this limitation, our healthy control data align closely with established normal ranges; therefore, we consider the data derived from healthy controls to be reliable. Considering the seasonal nature of scrub typhus and the fact that our case number exceeds that of previous studies, we believe that our data offer significant value. Second, disparities in age between the control group and patients with scrub typhus were observed. While age alone did not account for the significant differences in ANA titer between groups, achieving a more uniform age distribution across these groups probably would have provided clearer insights into the variations in ANA and anti-dsDNA IgM levels.

In conclusion, this study demonstrated a significant elevation in ANA and anti-dsDNA IgM titers in patients with scrub typhus compared to healthy controls, suggesting an autoimmunological response associated with scrub typhus infection. The progression of scrub typhus is marked by an increased intensity of immune responses, as evidenced by the temporal variation in antibody titers following symptom onset. These findings underscore the potential of infections like scrub typhus to trigger or exacerbate autoimmune responses, offering insights into the complex interplay between infectious diseases and autoimmunity. This research contributes to public health by improving our understanding of the potential connections between infectious diseases and autoimmune responses. Recognizing scrub typhus as a possible trigger for autoimmune conditions provides valuable insights that may inform the development of better diagnostic and therapeutic strategies. By identifying these links, we aim to enhance clinical outcomes for patients by enabling earlier interventions and more targeted treatments for both infectious diseases and autoimmune disorders.

## Supporting information

**S1 Table. Subgroup Analysis of Demographic and Immunological Characteristics of Healthy Controls and Scrub Typhus Patients Aged 40–59 Years.**
(DOCX)

**S2 Table. Logistic Regression Analysis of Factors Associated with Low Serum Albumin Levels (≤ 3.5 g/dL) in Patients with Scrub Typhus.**
(DOCX)

**S3 Table. Logistic Regression Analysis of Factors Associated with Low C4 Levels (≤ 40 mg/dL) in Patients with Scrub Typhus.**
(DOCX)

**S4 Table. Logistic Regression Analysis of Factors Associated with high anti-dsDNA IgM Levels (≧ 15 IU/mL) in Patients with Scrub Typhus.**
(DOCX)

**S5 Table. Logistic Regression Analysis of Factors Associated with high hs-CRP Levels (≧ 100 mg/L) in Patients with Scrub Typhus.**
(DOCX)

**S6 Table. Proportion of Anti-dsDNA IgM Positivity (≥ 15 U/mL) According to ANA Titer Levels in Patients with Scrub Typhus.**
(DOCX)

## Author contributions

**Conceptualization:** Hwanseung You, Yunjung Choi, Chang-Seop Lee.

**Data curation:** Hwanseung You, Jeongeun Song, Seonglyeong Kim, Yunjung Choi, Chang-Seop Lee.

**Formal analysis:** Hwanseung You, Sang-Min Oh, Joo-Hee Hwang, Yunjung Choi, Chang-Seop Lee.

**Investigation:** Hwanseung You, Sang-Min Oh, Joo-Hee Hwang, Jeong-Hwan Hwang.

**Methodology:** Hwanseung You, Jeongeun Song, Seonglyeong Kim.

**Supervision:** Wan-Hee Yoo, Yunjung Choi, Chang-Seop Lee.

**Validation:** Wan-Hee Yoo, Yunjung Choi, Chang-Seop Lee.

**Writing – original draft:** Hwanseung You, Yunjung Choi, Chang-Seop Lee.

**Writing – review & editing:** Hwanseung You, Jeongeun Song, Seonglyeong Kim, Sang-Min Oh, Joo-Hee Hwang, Jeong-Hwan Hwang, Wan-Hee Yoo, Yunjung Choi, Chang-Seop Lee.

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
