## [Decision Letter · Decision Letter 0]

19 Jun 2024

Dear Dr. Lee,

Thank you very much for submitting your manuscript "Scrub Typhus Association with Autoimmune Biomarkers and Clinical Implications" for consideration at PLOS Neglected Tropical Diseases. As with all papers reviewed by the journal, your manuscript was reviewed by members of the editorial board and by several independent reviewers. In light of the reviews (below this email), we would like to invite the resubmission of a significantly-revised version that takes into account the reviewers' comments. 

Please respond to the reviewers' comments.

Please explain whether the age differences between groups affect the results.

We cannot make any decision about publication until we have seen the revised manuscript and your response to the reviewers' comments. Your revised manuscript is also likely to be sent to reviewers for further evaluation.

Sincerely,

Yong Qi

Academic Editor

Mathieu Picardeau

Section Editor

Please respond to the reviewers' comments.

Please explain whether the age differences between groups affect the results.

Reviewer's Responses to Questions

**Key Review Criteria Required for Acceptance?**

**Methods**

-Are the objectives of the study clearly articulated with a clear testable hypothesis stated?

-Is the study design appropriate to address the stated objectives?

-Is the population clearly described and appropriate for the hypothesis being tested?

-Is the sample size sufficient to ensure adequate power to address the hypothesis being tested?

-Were correct statistical analysis used to support conclusions?

-Are there concerns about ethical or regulatory requirements being met?

Reviewer #1: (No Response)

Reviewer #2: 1. There is a significant difference in the age distribution between your control group and the scrub typhus group. You might need to address this in the methods section to enhance the credibility of the changes in autoantibodies after infection, which is the primary outcome of the article. This is because the proportion of ANA exceeding 1:80 is inherently higher in older individuals. Or you may consider the subgroup analyses based on age

2. Since the study examines temporal variation in antibody titers, specifying the follow-up duration and intervals for patient monitoring in methods would provide clearer insights into the longitudinal aspect of the study.

3. May consider using multivariate regression analysis to control for confounding factors and more clearly determine the independent relationship between ANA titers and other clinical and laboratory parameters.

Reviewer #3: objective of the study is not articulated .indication of doing ANA titre and anti DS DNA in a diagnosed case of scrub typhus is not clear.

**Results**

-Does the analysis presented match the analysis plan?

-Are the results clearly and completely presented?

-Are the figures (Tables, Images) of sufficient quality for clarity?

Reviewer #1: (No Response)

Reviewer #2: 1. The difference of age group with the trend of ANA titer in Table 3 indicate the autoantibodies being affected by age.

2. I suggest breaking down the proportions of several important ANA patterns, especially in cases where ANA > 1:80, to clarify whether there are patterns consistent with ds-DNA.

Reviewer #3: yes

**Conclusions**

-Are the conclusions supported by the data presented?

-Are the limitations of analysis clearly described?

-Do the authors discuss how these data can be helpful to advance our understanding of the topic under study?

-Is public health relevance addressed?

Reviewer #1: (No Response)

Reviewer #2: 1. As my opinion, ANA is an important screening tool and entry criterion for diagnosing SLE, with high sensitivity, low specificity.

2. You may need focus on the potential pathogenesis why scrub typhus infection trigger early pathologic course of SLE due to you try to enhance the link between them.

Reviewer #3: yes

**Editorial and Data Presentation Modifications?**

Reviewer #1: (No Response)

Reviewer #2: (No Response)

Reviewer #3: minor revision. objective of the study is not articulated .indication of doing ANA titre and anti DS DNA in a diagnosed case of scrub typhus is not clear. this can be added.

**Summary and General Comments**

Reviewer #1: (No Response)

Reviewer #2: (No Response)

Reviewer #3: minor revision. objective of the study is not articulated .indication of doing ANA titre and anti DS DNA in a diagnosed case of scrub typhus is not clear. this can be added.

PLOS authors have the option to publish the peer review history of their article (what does this mean? ). If published, this will include your full peer review and any attached files.

**Do you want your identity to be public for this peer review?** For information about this choice, including consent withdrawal, please see our Privacy Policy .

Reviewer #1: No

Reviewer #2: No

Reviewer #3: Yes: Minakshi Rohilla
---

## [Decision Letter · Decision Letter 1]

25 Aug 2024

Dear Dr. Lee,

Thank you very much for submitting your manuscript "Scrub Typhus Association with Autoimmune Biomarkers and Clinical Implications" for consideration at PLOS Neglected Tropical Diseases. As with all papers reviewed by the journal, your manuscript was reviewed by members of the editorial board and by several independent reviewers. In light of the reviews (below this email), we would like to invite the resubmission of a significantly-revised version that takes into account the reviewers' comments. 

We cannot make any decision about publication until we have seen the revised manuscript and your response to the reviewers' comments. Your revised manuscript is also likely to be sent to reviewers for further evaluation.

Sincerely,

Yong Qi

Academic Editor

Joseph Vinetz

Section Editor

Reviewer's Responses to Questions

**Key Review Criteria Required for Acceptance?**

**Methods**

-Are the objectives of the study clearly articulated with a clear testable hypothesis stated?

-Is the study design appropriate to address the stated objectives?

-Is the population clearly described and appropriate for the hypothesis being tested?

-Is the sample size sufficient to ensure adequate power to address the hypothesis being tested?

-Were correct statistical analysis used to support conclusions?

-Are there concerns about ethical or regulatory requirements being met?

Reviewer #1: The revised methods section was appropriately addressed in the revised version. There are no additional concerns.

Reviewer #3: 1. It is still not clear about the retrospective analysis of the lab reports whether it was serial testing of the samples kept in the past OR we are just analyzing the reports of serial test done in the past.

2. if we are just analyzing the reports of serial test done in the past. what was the indication of doing autoimmune workup in the past at the time of scrub typhus infection 

please clarify this in the methods.

Reviewer #4: The authors have presented four different methodologies for assigning scrub typhus positivity. Criteria 1-3 involve serological methods and criteria for uses PCR. what is the evidence for the use of criteria 1 and 2? Has this methodology been validated in a endemic area? How do you know that it is not residual antibodies from a recent or previous infection? Scrub typhus for antibodies can persist for months and years especially IgG. Please provide conclusive evidence for the assignment of scrub typhus positivity using these diagnostic criteria.

Please provide more information regarding the IFA test that was performed. What were the strains used, how many readers, etc? This is not provided in the manuscript at the moment.

Reviewer #5: Good Work. The objectives and and methodology is described properly.

Reviewer #6: Please see the comments in the Results Section.

**Results**

-Does the analysis presented match the analysis plan?

-Are the results clearly and completely presented?

-Are the figures (Tables, Images) of sufficient quality for clarity?

Reviewer #1: The authors carefully addressed the comments from the reviewers and corrected text accordingly. The authors still need to provide figures in better quality. The ones provided in the revised manuscript are still of poor quality. There is no additional concern regarding the analysis.

Reviewer #3: not clear

Reviewer #4: Furthermore, please provide a table that demonstrates positivity for each of the criteria. Presumably, there would be some patients that were positive for multiple criteria? This needs to be shown in a table to demonstrate the strength of evidence for true scrub typhus infection. It would be very interesting to see what is the percentage of serology positives compared to PCR positives.

Reviewer #5: The analysis presented match the analysis plan.The results clearly and completely presented.

Reviewer #6: 1. Antibody Type and Clinical Manifestations: It is acknowledged that the major pathogenetic type of anti-dsDNA in lupus is IgG rather than IgM, this study does not delve into the relationship between IgG levels and specific clinical manifestations. This omission limits the depth of the research and fails to adequately clarify the clinical significance of IgG.

2.ANA Pattern Analysis: The ANA pattern in both scrub typhus patients and controls should be analyzed. And it does not further analyze the clinical relevance of this finding, which weakens the interpretative power of the results.

3.Association of ANA Positivity with Clinical manifestations: The relationship between ANA positivity and clinical manifestations should be further analyzed to confirm that scrub typhus infection may be associated with immune-induced vasculitis. 

4.Errors in the Manuscript: The manuscript contains several minor errors, such as the disorganization of figures, which affects the readability and understanding of the results.

**Conclusions**

-Are the conclusions supported by the data presented?

-Are the limitations of analysis clearly described?

-Do the authors discuss how these data can be helpful to advance our understanding of the topic under study?

-Is public health relevance addressed?

Reviewer #1: The conclusion and discussion sections were carefully revised by the authors based on comments from reviewers. There are no additional concerns regarding the conclusion.

Reviewer #3: (No Response)

Reviewer #4: It's not clear what is the purpose of the study in the context of scrub typhus disease. Therefore there needs to be some statement at the start of the manuscript to the importance of the disease and then have some type of conclusion relating to the aims and objectives of the study. What is the public health relevance of this finding, etc.

Reviewer #5: The conclusions are supported by the data presented

Reviewer #6: 1.The article lacks innovation in its research content, particularly in the comparative analysis of ds-DNA and ANA levels produced during infections. It fails to provide significant clinical value for the diagnosis and treatment of scrub typhus, resulting in limited scientific contribution.

2.The mechanisms linking infection and vasculitis are complex. During infections, many low-affinity antibodies (such as anti-dsDNA and antiphospholipid antibodies) are produced, which may contribute to small vessel inflammation. Additionally, innate immune cells, such as neutrophils, can directly damage endothelial cells, while inflammatory factors like interferons can activate immune cells to produce cytokines that also harm endothelial cells. However, the study does not provide a detailed discussion of the mechanisms by which these antibodies and immune cells influence inflammatory responses, limiting its comprehensiveness.

**Editorial and Data Presentation Modifications?**

Reviewer #1: Figures need to be reproduced in high quality.

Reviewer #3: minor revision

Reviewer #4: The discussion is a bit pedestrian and could be tightened up.

Reviewer #5: May be accepted.

Reviewer #6: (No Response)

**Summary and General Comments**

Reviewer #1: (No Response)

Reviewer #3: 1. It is still not clear about the retrospective analysis of the lab reports whether it was serial testing of the samples kept in the past OR we are just analyzing the reports of serial laboratory test done in the past.

2. if we are just analyzing the reports of serial laboratory test done in the past. what was the indication of doing autoimmune workup in the past at the time of scrub typhus infection 

please clarify this in the methods.

Reviewer #4: It's not clear what is the purpose of this study. I do understand that the study of antinuclear antibodies is interesting, but what's the purpose in this context?

What is the purpose of multi sequence typing in this context. It doesn't seem to make any sense given that the study is focusing on ANA?

Reviewer #5: Overally the novelty is clear and the paper is organised properly.

Reviewer #6: This article analyzes the levels of ANA and anti-dsDNA-IgM in patients with scrub typhus infection, aiming to elucidate the relationship between scrub typhus infection and autoimmunity. I believe the following issues need to be considered.

1.The article lacks innovation in its research content, particularly in the comparative analysis of ds-DNA and ANA levels produced during infections. It fails to provide significant clinical value for the diagnosis and treatment of scrub typhus, resulting in limited scientific contribution.

2.Antibody Type and Clinical Manifestations: It is acknowledged that the major pathogenetic type of anti-dsDNA in lupus is IgG rather than IgM, this study does not delve into the relationship between IgG levels and specific clinical manifestations. This omission limits the depth of the research and fails to adequately clarify the clinical significance of IgG.

3.ANA Pattern Analysis: The ANA pattern in both scrub typhus patients and controls should be analyzed. And it does not further analyze the clinical relevance of this finding, which weakens the interpretative power of the results.

4.Association of ANA Positivity with Clinical manifestations: The relationship between ANA positivity and clinical manifestations should be further analyzed to confirm that scrub typhus infection may be associated with immune-induced vasculitis. 

5.The mechanisms linking infection and vasculitis are complex. During infections, many low-affinity antibodies (such as anti-dsDNA and antiphospholipid antibodies) are produced, which may contribute to small vessel inflammation. Additionally, innate immune cells, such as neutrophils, can directly damage endothelial cells, while inflammatory factors like interferons can activate immune cells to produce cytokines that also harm endothelial cells. However, the study does not provide a detailed discussion of the mechanisms by which these antibodies and immune cells influence inflammatory responses, limiting its comprehensiveness.

6.Errors in the Manuscript: The manuscript contains several minor errors, such as the disorganization of figures, which affects the readability and understanding of the results.

PLOS authors have the option to publish the peer review history of their article (what does this mean? ). If published, this will include your full peer review and any attached files.

**Do you want your identity to be public for this peer review?** For information about this choice, including consent withdrawal, please see our Privacy Policy .

Reviewer #1: No

Reviewer #3: Yes: Minakshi Rohilla

Reviewer #4: No

Reviewer #5: No

Reviewer #6: No
---

## [Decision Letter · Decision Letter 2]

6 Dec 2024

Dear Dr. Lee,

We are pleased to inform you that your manuscript 'Scrub Typhus Association with Autoimmune Biomarkers and Clinical Implications' has been provisionally accepted for publication in PLOS Neglected Tropical Diseases.

Best regards,

Yong Qi

Academic Editor

Joseph Vinetz

Section Editor

Shaden Kamhawi

co-Editor-in-Chief

Paul Brindley

co-Editor-in-Chief

Reviewer's Responses to Questions

**Key Review Criteria Required for Acceptance?**

**Methods**

-Are the objectives of the study clearly articulated with a clear testable hypothesis stated?

-Is the study design appropriate to address the stated objectives?

-Is the population clearly described and appropriate for the hypothesis being tested?

-Is the sample size sufficient to ensure adequate power to address the hypothesis being tested?

-Were correct statistical analysis used to support conclusions?

-Are there concerns about ethical or regulatory requirements being met?

Reviewer #4: The methodological revisions have addressed age distribution concerns through age stratification, subgroup analysis, and statistical adjustments, ensuring that observed ANA titer changes are due to scrub typhus rather than age. Additionally, the authors' detailed temporal analysis of antibody titers and clinical parameters over the disease course strengthens the evidence for scrub typhus triggering or mimicking autoimmune responses.

**Results**

-Does the analysis presented match the analysis plan?

-Are the results clearly and completely presented?

-Are the figures (Tables, Images) of sufficient quality for clarity?

Reviewer #4: The revisions to the results section include a more detailed analysis of the temporal progression of ANA and anti-dsDNA IgM titers in scrub typhus patients, highlighting significant increases over time compared to healthy controls. The authors also expanded the comparison of clinical parameters, such as complement C4 and serum albumin, finding that higher ANA titers were associated with a decline in C4 and elevated anti-dsDNA IgM levels.

**Conclusions**

-Are the conclusions supported by the data presented?

-Are the limitations of analysis clearly described?

-Do the authors discuss how these data can be helpful to advance our understanding of the topic under study?

-Is public health relevance addressed?

Reviewer #4: The discussion section revisions emphasise the implications of the study's findings on the potential autoimmune effects of scrub typhus, now strengthened by temporal data on antibody titers and clinical markers. The authors draw clearer parallels between scrub typhus and autoimmune conditions like SLE, suggesting that the infection could serve as a trigger for autoimmunity in predisposed individuals. They also address the limitations more thoroughly, particularly the impact of age and the seasonal nature of scrub typhus, which reinforces the study’s scope and validity.

**Editorial and Data Presentation Modifications?**

Reviewer #4: None

**Summary and General Comments**

Reviewer #4: None

PLOS authors have the option to publish the peer review history of their article (what does this mean? ). If published, this will include your full peer review and any attached files.

**Do you want your identity to be public for this peer review?** For information about this choice, including consent withdrawal, please see our Privacy Policy .

Reviewer #4: No

---

## [Editor Report · Acceptance letter]

Dear Dr. Lee,

We are delighted to inform you that your manuscript, "Scrub Typhus Association with Autoimmune Biomarkers and Clinical Implications," has been formally accepted for publication in PLOS Neglected Tropical Diseases.

Best regards,

Shaden Kamhawi

co-Editor-in-Chief

Paul Brindley

co-Editor-in-Chief
